# Exploring the impacts of the 2012 Health and Social Care Act reforms to commissioning on clinical activity in the English NHS: a mixed methods study of cervical screening

Jonathan Hammond,[1,2,3] Thomas Mason,[1,2,3,4] Matt Sutton,[1,2,3] Alex Hall,[2,5] Nicholas Mays,[6] Anna Coleman,[1,2,3] Pauline Allen,[7] Lynsey Warwick-Giles,[1,2,3] Kath Checkland[1,2,3]

For numbered affiliations see end of article.

**Correspondence to**
Dr Jonathan Hammond;
jonathan.hammond@
manchester.ac.uk

## ABSTRACT

**Objectives** Explore the impact of changes to commissioning introduced in England by the Health and Social Care Act 2012 (HSCA) on cervical screening activity in areas identified empirically as particularly affected organisationally by the reforms.

**Methods** Qualitative followed by quantitative methods. Qualitative: semi-structured interviews (with NHS commissioners, managers, clinicians, senior administrative staff from Clinical Commissioning Groups (CCGs), local authorities, service providers), observations of commissioning meetings in two metropolitan areas of England. Quantitative: triple-difference analysis of national administrative data. Variability in the expected effects of HSCA on commissioning was measured by comparing CCGs working with one local authority with CCGs working with multiple local authorities. To control for unmeasured confounders, differential changes over time in cervical screening rates (among women, 25–64 years) between CCGs more and less likely to have been affected by HSCA commissioning organisational change were compared with another outcome—unassisted birth rates—largely unaffected by HSCA changes.

**Results** Interviewees identified that cervical screening commissioning and provision was more complex and 'fragmented', with responsibilities less certain, following the HSCA. Interviewees predicted this would reduce cervical screening rates in some areas more than others. Quantitative findings supported these predictions. Areas where CCGs dealt with multiple local authorities experienced a larger decline in cervical screening rates (1.4%) than those dealing with one local authority (1.0%). Over the same period, unassisted deliveries decreased by 1.6% and 2.0%, respectively, in the two groups.

**Conclusions** Arrangements for commissioning and delivering cervical screening were disrupted and made more complex by the HSCA. Areas most affected saw a greater decline in screening rates than others. The fact that this was identified qualitatively and then confirmed quantitatively strengthens this finding. The study suggests large-scale health system reforms may have unintended consequences, and that complex commissioning arrangements may be problematic.

---

### Strengths and limitations of this study

► Few studies have investigated in detail the impacts of large-scale health system change.
► This study combines detailed qualitative data exploring impacts on the system with quantitative exploration of important outcomes, supporting causal inference.
► Based on qualitative findings, we developed a quantitative measure for assessing the extent of disruption to the English NHS commissioning system as a result of the 2012 Health and Social Care Act.
► We found that cervical screening rates decreased more post-Act in areas that had experienced higher levels of disruption.

---

## INTRODUCTION

Structural reorganisations of publicly financed healthcare systems, driven by central government or other state agencies, are frequently employed with the objective of improving healthcare delivery, and thus population health outcomes, while reducing or containing costs.[1] However, such endeavours can be disruptive and expensive.[2] It is important to understand what possible impacts these reorganisations have in order to understand their value.[3]

In the English National Health Service (NHS), attempts to evaluate the impact of reorganisations have typically used operational indicators (eg, bed availability, number of staff) and measures of clinical activity because their improvement was the stated goal of government policy (eg, The NHS Plan[4]). Other studies have attempted to assess the impacts of reforms by measuring their effects on prices, quality and quantity of provision.[5] Most studies have relied

on quantitative analysis of measures that were explicitly targeted by policy reforms. There is a need for approaches which combine qualitative and quantitative methods to generate a deeper understanding of the impacts of structural reorganisation.[6]

The most recent structural reorganisation of the English NHS, the Health and Social Care Act[7] (hereafter 'HSCA' or 'the Act'), was introduced in April 2013 and included wide-ranging changes to the health services commissioning system. We explore whether changes to the commissioning of cervical screening services resulting from the Act affected uptake. This analysis uses a relatively novel mixed methods approach. An initial 'bottom-up' qualitative analysis allowed us to identify problematic issues associated with the HSCA for those working locally in the health service commissioning system. This process highlighted the disruption to established commissioning arrangements and cervical screening as a clinical activity, which may be specifically affected by this disruption. We then developed a quantitative investigation to explore this more fully. Together these analyses allow us to infer causation.

### The HSCA and changes to cervical screening commissioning

The HSCA is regarded as one of the most wide-ranging legislative reforms in the history of the English NHS.[8] Primary Care Trusts (PCTs), 152 organisations previously responsible for the commissioning of primary, community and secondary health services from providers on behalf of local populations, were abolished. Their commissioning functions were split between three groups of organisations: 211 (now 195) newly created Clinical Commissioning Groups (CCGs), membership organisations constituted by general practitioner (GP) (family doctor) practices, given responsibility for commissioning services for their local populations; NHS England (NHSE), a new arm's-length governmental body with responsibility for authorising and assessing CCGs and commissioning some services at a national level; and top-tier and single-tier elected local authorities, which took responsibility for the majority of public health services for the first time since 1974. In addition, Public Health England (PHE) was created as an executive agency of the Department of Health, to unify the diverse public health profession and provide expert support for local public health services.

In some service areas, the transfer of commissioning responsibilities was relatively straightforward (eg, the commissioning of routine orthopaedic surgery was passed from PCTs to CCGs with minimal alteration to the bundle of services involved). In other service areas, the transfers were much more complex, particularly the commissioning of national screening programmes and sexual health services, as a result of changes to public health commissioning. Pre-HSCA, national screening programmes and sexual health services were both commissioned by PCTs. Cervical screening was largely provided by GP practices, which received additional funding linked to levels of activity,[9] but some women opted to have their cervical

smears in PCT-commissioned sexual health clinics. Post-HSCA, responsibility for public health services, including most sexual health services, was transferred to local authorities. The underlying programme theory (ie, the explicit expectation about how the policy would work[10]) was that local authorities would be better placed to address the wider determinants of health and well-being than the NHS because they could link public health services with their existing responsibilities, such as for transport and housing.[11] NHSE took responsibility for commissioning national screening programmes.[8] NHSE's regional teams are responsible for commissioning screening programmes, supported by PHE staff 'embedded' within NHSE's screening and immunisation teams.[12] There was no identifiable underlying programme theory for this specific change to screening programme commissioning. However, it is notable that, in contrast to the emphasis placed on localism associated with the creation of CCGs and with the transfer of public health to local authorities, screening commissioning became more centralised as a consequence of the HSCA.

Table 1 shows the organisations with responsibilities of relevance to the commissioning of sexual health, including cervical cancer screening services, pre-HSCA and post-HSCA. It illustrates how responsibility for such services, previously commissioned by PCTs, was split between different agencies. This increased complexity and fragmentation of responsibilities had the potential to disrupt service commissioning.[13]

This analysis comes from a study designed to foster emergent interplay between qualitative and quantitative data analysis.[14 15] The focus on cervical screening was not established at the project design stage but driven by the initial qualitative interview findings related to sexual health commissioning arrangements and screening activity post-HSCA. These findings prompted us to consider a quantitative exploration of predictions made by interviewees relating to potential changes in cervical screening activity.

## METHODS

### Study context and design

This analysis forms part of a longitudinal project, with data collected between January 2015 and December 2017, into the effect of the HSCA on the commissioning system in England. We combined a qualitative and quantitative exploration of the commissioning of services in two large, socioeconomically diverse metropolitan areas of England with a national level quantitative study of commissioning outcomes. We used a sequential mixed methods approach in which initial qualitative data collection and analysis were used to shape an ensuing quantitative analysis using routinely available data. We therefore present the qualitative and quantitative methods and findings in the order undertaken, and integrate them in the 'Discussion' section.

**Table 1** Organisations of significance to the commissioning of cervical screening pre-HSCA and post-HSCA

| Pre-HSCA | |
|---|---|
| Primary Care Trusts | Responsible for all public health commissioning, including sexual health services; responsible for commissioning national screening programmes, including cervical screening. |
| UK National Screening Committee | Provision of advice and support to NHS organisations on population screening programmes. |
| **Post-HSCA** | |
| Local authorities | Responsible for most public health commissioning, including most sexual health services (contraception over and above GP contract, testing and treatment of sexually transmitted infections, sexual health advice and promotion). |
| NHS England (NHSE) | Responsible for commissioning national screening programmes, including cervical screening, and some sexual health services (notably contraception through GP contract, and HIV treatment). |
| Public Health England (PHE) (including National Screening Committee) | Hosts UK National Screening Committee secretariat, which retains the same role; PHE staff embedded in subregional NHSE screening and immunisation teams to provide expertise and 'leadership'. |
| Clinical Commissioning Groups (CCGs) | Responsible for improving clinical outcomes for their patients and prevent premature death (as part of the NHS Outcomes Framework, which CCGs are held to account on by NHSE); GP members are funded to carry out cervical screening through the GP contract. |

GP, general practitioner; HSCA, Health and Social Care Act 2012.

## Patient and public involvement

Our interest in exploring the impact of systemic commissioning change on cervical screening rates was driven initially by concerns expressed by interviewees about potentially negative consequences for patients relating to new arrangements. Patients were not directly involved in the design of, or recruitment to, the overarching project, but an advisory group including a patient representative met regularly throughout the project and played an important role in supporting its development. We presented our initial qualitative findings relating to cervical screening, and early ideas for developing a mixed methods investigation, to our advisory group and were encouraged by our patient representative to pursue this. The results of the broader project were disseminated to participants, and the advisory group, in the form of a series of short reports focusing on specific areas of commissioning and the final report.

## Qualitative component
### Setting, participants, sampling and data collection

The qualitative component took place between March 2015 and August 2017, focussing on two metropolitan 'health economies' covering a geographical population and a group of commissioning organisations and providers with close operational links. Across both areas, we conducted 143 interviews (each typically an hour in length), 93 of which related to sexual health commissioning, with clinical and non-clinical commissioners, managers, clinicians and senior administrative staff from CCGs, local authorities, service providers and third sector organisations. Organisations and participants were sampled purposively for variation in type and role. We identified participants through organisational websites, personal contacts and through 'snowballing' in which we asked participants to recommend other potential participants. Additionally, 8 hours of meetings of an interorganisational sexual health coordinating group involving sexual health commissioners and providers were observed in one of the areas.

Interviews focused on the commissioning system pre-HSCA and post-HSCA, exploring continuities and changes to personal and organisational roles, key issues and challenges, accountability and performance management, interorganisational relationships and communication and commissioning decision-making. Interviews took place either in person (usually in participants' offices) or over the telephone. All interviews were audio-recorded and all interviewees were provided with written information about the study before consenting to participate.

### Data analysis

Audio recordings of interviews were transcribed verbatim, and observational notes from meetings were produced contemporaneously. Transcripts and observational notes were imported into NVivo V.10 software and analysed thematically by JH and AH.[16] This involved repeated readings of transcripts to become sufficiently familiar with their contents, identifying initial codes and coding chunks of data, searching for themes and then iteratively defining and reconstituting themes. Our findings (see below) contained some predictions made by participants regarding changes in cervical screening activity as a consequence of the Act. This prompted us to explore these predictions in a quantitative analysis, which we now describe.

### Quantitative component

Interviewees identified that the HSCA had introduced confusion over responsibility for the commissioning of cervical screening services, and had increased variability of provision. They hypothesised that cervical screening rates might be reduced by the new commissioning arrangements; this prompted discussions among the research team and advisory group about developing a way of testing these predictions quantitatively as far as routine data would permit. As the Act had been implemented in all areas simultaneously, removing the scope for a quasi-experimental approach, we sought to identify a measure of variability in the extent to which the Act would have been expected to make commissioning more difficult in each area. One of the features of the post-HSCA system was that some, but not all, CCGs were established which crossed local authority boundaries. Some CCGs related to as many as three separate local authorities. Local authorities were now directly involved in sexual health services commissioning. Findings revealed that CCGs experienced extra challenges when they had to engage with more than one local authority. This suggested that the burden of additional interorganisational coordination might have consequences for commissioning.

As each local authority developed its own approach to cervical screening in its local sexual health clinics, we explored the possibility that GP practices located in CCGs which had to work with more than one local authority might experience lower screening rates compared with practices located in CCGs which had only to deal with one local authority. We compare the demographic characteristics of these two groups in table 2. The 89 CCGs dealing with more than one local authority had a slightly older population profile than the 119 CCGs which dealt with only one local authority but were otherwise highly comparable.

Because cervical screening rates may be influenced by other factors that we cannot observe and may change over time in different ways between the two groups of CCGs, we also compared screening rates with an outcome that was unlikely to have been affected by the introduction of the HSCA. We used unassisted births (ie, uncomplicated deliveries which did not require any intervention) as a percentage of all maternal deliveries, since the commissioning of maternity services was largely unchanged by the Act.

We applied a triple-difference approach. The triple difference represents (the change over time in cervical screening rates for CCGs working with only one local authority minus the change over time in cervical screening rates for CCGs working with more than one local authority) minus (the change over time in unassisted birth rates for CCGs working with only one local authority minus the change over time in unassisted birth rates for CCGs working with more than one local authority).

The screening rate is defined as the percentage of women aged between 25 and 64 years who had received a cervical screening test in the preceding 5 years. This indicator was derived from annual, practice-level data from the Quality and Outcomes Framework, 2009–10 to 2015–16. The comparison indicator is unassisted births as a percentage of all maternal deliveries. This indicator was produced using operation codes in Hospital Episode Statistics for 2009–10 to 2015–16. We aggregated the spell-level data by general practice and financial year.

The key assumption underpinning the triple-difference estimator is that, conditional on the other variables in the model, the differences in the changes over time in the intervention indicator (cervical screening) between the 'exposed' and the 'control' areas (in this case, CCGs working with one local authority vs CCGs working with more than one) would have been the same as the differences in the changes over time in the comparison indicator (unassisted births) between the exposed and control areas in the absence of the intervention. This is a more complex version of the 'parallel trends' assumption required for the double-difference, or difference-in-differences, estimator.[17]

A popular test of this assumption in the double-difference case is that there are parallel trends over time in the outcomes in the intervention and comparison group in the preintervention period. For our triple-difference case, we used an F-test to assess the joint significance of interactions between the year effects and the binary variable representing the combination of exposed area and treated indicator in the preperiod.

We also used the lagged dependent variable (LDV) estimator. This model is estimated only on data in the postintervention period and is a less biased estimator of treatment effects when the assumption of parallel pretrends does not hold.[18] We set up the LDV model to generate the equivalent impact estimate as the triple-difference model. The model included: dummy variables for year; values of the dependent variable in each of the preintervention periods; a dummy variable classifying practices depending on whether they were located in

**Table 2** Clinical Commissioning Group (CCG) demographic characteristics depending on the number of local authorities that the CCG needs to work with

| | One local authority | More than one local authority |
|---|---|---|
| Number of CCGs | 119 | 89 |
| Population (millions) | 29.5 | 26.9 |
| Female | 50.0% | 50.5% |
| Age (years) | | |
| 0–9 | 12.3% | 11.4% |
| 10–19 | 11.2% | 11.2% |
| 20–39 | 30.0% | 24.8% |
| 40–59 | 26.5% | 27.7% |
| 60–79 | 16.0% | 19.7% |
| 80 and above | 4.1% | 5.3% |

CCGs working with more than one local authority; interactions between year and condition dummies; interactions between values of the dependent variable in the preintervention period and the condition dummy and an interaction between the dummy variable classifying practices depending on whether they were located in CCGs working with more than one local authority and the condition dummy. The final term is the impact estimate, showing whether cervical screening was differentially affected after the introduction of the reforms for local authorities working with multiple CCGs.

We estimated the regression models in Stata V.14.1 using dummy variable weighted least squares regression with fixed effects for practice-indicator combinations. The ways in which these models are estimated using regression analyses are described formally in the technical online supplementary appendix. As the dependent variable is a proportion, and constrained to lie between 0 and 1, we used the empirical logit transformation and back-transformed the coefficients and associated 95% CIs using the mean value of the cervical screening rate.[19] We clustered the SEs at the GP practice level.[20] The general form of the STATA command is: *areg {depvar} {indepvars} [aw=denom], robust absorb(practicexindicator) cluster(practice).*

## RESULTS
### Qualitative findings
Interviewees told us that CCGs working with more than one local authority experienced a number of challenges, including: finding sufficient capacity to engage in multiple meetings of the same type with different local authorities; managing additional collaborative relationships; working with organisations experiencing different financial pressures from each other with different approaches to public health spending; and attempting to develop integrated health and social care arrangements with one local authority that did not have unintended and undesirable consequences for plans with another. The following extract illustrates issues relating to difficulties commissioning a single service offer for CCG patients and the additional resources required for a CCG working with multiple local authorities. (Interview data extracts are denoted by square brackets with numerical participant ID, participant's organisation type, Area (1 or 2) and month and year of the interview.)

> We do have two sets of safeguarding arrangements. So I guess at one level, one can say there is a risk of and there are examples of services being subtly different. Equally, you've got to service two times the number of these processes, which can be quite labour-intensive. [2778, CCG, Area 1, April 2015]

In our analysis relating directly to issues surrounding the commissioning and provision of cervical screening post-HSCA, we identified two main themes: confusion and uncertainty regarding budgets and responsibilities, and potential impacts on cervical screening rates. Many of the issues discussed below are likely to be exacerbated when the number of interacting commissioning organisations in a local area are increased.

### Confusion and uncertainty regarding budgets and responsibilities
Before the HSCA, both cervical screening and sexual health services were commissioned by PCTs. As one screening and immunisation lead outlined, cervical screening tests (sometimes referred to as smear tests) were provided by GP practices, but patients could usually also have them at sexual health clinics [17685, NHSE, Area 2, December 2016]. Whereas pre-HSCA PCTs held the budget for both cervical screening and sexual health services, following the Act these budgets were separated. This meant that the local authority budget and responsibility for sexual health did not extend to cervical screening. One local authority public health consultant reported that, in spite of this, PHE was sending letters to patients explicitly stating that they could choose to attend either their GP practice or their local sexual health clinic for their cervical screening test. This highlights confusion regarding commissioning arrangements and budgetary responsibility:

> Public Health England were writing around to people saying …you're due your smear, you can go to your general practice or you can go to your local sexual health clinic. And we said, but we don't have the money for them to do that, they can't come here routinely unless you're going to pay us for that. Public Health England, the screening people, they have the money to pay for the smears. But in all the moving around of the budgets, the money for smears that were taken outside general practice doesn't seem to be anywhere. [8384, local authority, Area 1, November 2015]

One participant from NHSE offered a different perspective. He argued that the public health budget of each local authority reflected the levels of cervical screening activity that had taken place in its sexual health clinics pre-HSCA. However, this is not clear because, in the past, the funding was not 'disaggregated' [4058, NHSE, Area 1, June 2015]. Therefore, it is not possible to establish what the pre-HSCA sexual health component of the public health budget covered.

> …they (local authorities) think they're not being paid for it (cervical screening). But, actually, in truth, whatever they were doing at the point of transition if they were doing loads of cervical smears they were just doing loads of cervical smears, so they had the money. There wasn't a problem when they were doing them before, it's just the money wasn't disaggregated. However local authorities have been put under significant pressure in their public health teams to reduce their budgets. So these kinds of things are examples where you can say it's not our responsibility so therefore we're taking that element out. [4058, NHSE, Area 1, June 2015]

The above quote illustrates a phenomenon reported by a number of participants that local authorities had reprocured their sexual health services and had taken a position that they would not commission their sexual health provider(s) to do routine cervical screening, because it was not their commissioning responsibility. However, as one member of a screening team in Area 2 illustrated, NHSE was also reluctant to explicitly commission sexual health services to provide cervical screening, seemingly because of administrative challenges relating to numerous low-value contracts with providers:

> So cervical screening, we could go to every sexual health provider and have a separate contract. The difficulty again becomes around commissioning capacity. So, I think we've got [x] local authorities, so we have [x] separate contracts all very low value, it's about 1000 screens in each, so you're talking maybe [x] £20 000 contracts or something. So, it's a very bitty way of doing stuff. So, we could still do it and we could pay for it, but in terms of the amount of paperwork or the amount of outcomes it becomes potentially unmanageable. [17685, PHE/NHSE, Area 2, December 2016]

This participant went on to indicate that he would prefer local authorities to commission cervical screening as part of their sexual health contracts, but acknowledged the political difficulties for local authorities to justify spending money on an area of service that was not formally their responsibility, especially given the context of diminishing local authority budgets:

> In a way, wouldn't it be so much easier if the local authorities just included it as part of their normal service? But their argument would be that's not our role, and how can we defend to the (elected) councillors that we're spending money on stuff that we don't have to, that someone else is meant to be spending money on? And our argument is well, it's just so much simpler and it's not a lot of money. That's the kind of discussion. And it eventually ends up with them withdrawing money and us saying well, we're not buying it either then. [17685, PHE/NHSE, Area 2, December 2016]

### Potential impacts on cervical screening rates

One local authority commissioner suggested that the policy of his local authority was to continue to facilitate opportunistic cervical screening tests at sexual health clinics, but not routine tests, because to provide the latter would have a detrimental impact on other sexual health services that the local authority was now obligated to commission ("if we don't say no to (routine) smears, we'll be turning (other) people away, symptomatic patients away, or women needing contraception away. And that's our duty" [8384, local authority, Area 1, November 2015]). He reported that local CCGs complained about this discontinuation of routine cervical

screening at sexual health clinics, because there was insufficient capacity within general practice for CCGs to meet their cervical screening targets, and thus they required sexual health clinics to provide a proportion of cervical screening activity. One screening consultant developed this point by suggesting that some localities would see a substantial reduction in screening activity because of a lack of capacity within primary care:

> …in some local authorities where the sexual health service is no longer doing cervical screening (it) will have a small impact but not a huge impact, in other areas, it will have a big impact on coverage, we'll see activity go down around it, because the workload is just going to come straight back to primary care, and in different areas primary care didn't realise this was happening, the re-commissioning, hasn't got the capability and the capacity… [18352, PHE/NHSE, Area 1, January 2017]

Another screening consultant reflected that changes to NHSE 'footprints' (ie, the abolition of Area Teams and the new, more regional focus of the organisation) had implications for the provision of cervical screening:

> …say we wanted to sort out cervical screening coverage in GP practices, in (name of PCT) you've got [x] GP practices, bottom 20 per cent you could talk to the [y] practices. In my new patch we've got (many more than x) practices. So you have to think in a completely different way. [17685, NHSE, Area 2, December 2016]

Several participants from different localities in both geographical areas pointed to long-standing challenges in ensuring good uptake rates for screening among their diverse local populations. There were concerns that these challenges would be exacerbated by a reduction in choice for women about where they could go for cervical screening tests:

> …you should have an integrated sexual health service where predominantly women can go in and get seen in one episode, in one place for all their sexual health needs, be that sexually transmitted infection testing and treatment and contraception. So I think probably in the past people worked very hard to get things like cervical screening into these services so that the needs of those women who perhaps wouldn't go to their local GP could be met in an environment they felt happy with. My feeling is now… that perhaps the type of women who traditionally would have gone for cervical screening (at their sexual health clinic) might not feel so comfortable in that environment (of the GP practice). So particularly, say, a lady from a South Asian background who goes to a single handed male GP with no practice nurse, that's the kind of traditional person who might have gone to a family planning clinic for their cervical screening. [9742, local authority, Area 2, January 2016]

**Table 3** Numbers of general practices and mean rates of cervical screening and unassisted deliveries by year and by the number of LAs with which CCGs had to coordinate commissioning

| Year | Number of general practices | | | Cervical screening rate (%) | | | Unassisted delivery rate (%) | | |
|---|---|---|---|---|---|---|---|---|---|
| | 1 LA | 2+ LAs | All | 1 LA | 2+ LAs | All | 1 LA | 2+ LAs | All |
| 2009–10 | 4260 | 3399 | 7659 | 82.85 | 84.74 | 83.75 | 63.44 | 63.8 | 63.59 |
| 2010–11 | 4261 | 3403 | 7664 | 82.74 | 84.61 | 83.63 | 63.19 | 63.76 | 63.44 |
| 2011–12 | 4260 | 3400 | 7660 | 81.49 | 83.13 | 82.27 | 62.41 | 63.61 | 62.94 |
| 2012–13 | 4249 | 3399 | 7648 | 81.28 | 82.87 | 82.04 | 62.17 | 63.08 | 62.57 |
| 2013–14 | 4199 | 3355 | 7554 | 81.31 | 82.5 | 81.88 | 61.33 | 62.43 | 61.82 |
| 2014–15 | 4125 | 3302 | 7427 | 81.21 | 82.54 | 81.85 | 60.67 | 61.98 | 61.25 |
| 2015–16 | 4026 | 3236 | 7262 | 80.75 | 82.31 | 81.49 | 60.47 | 61.53 | 60.94 |
| Relative change between 2009–10 and 2015–16 (%) | | | | −2.53 | −2.87 | −2.70 | −4.68 | −3.56 | −4.17 |

Mean cervical screening and unassisted delivery rates are weighted by the denominators used in the calculation of the rates. These are the eligible populations; the number of women aged between 25 and 64 years or the number of maternal deliveries.
CCG, Clinical Commissioning Group; LA, local authority.

## Summary

The HSCA separated commissioning responsibilities for some types of services, including sexual health. Our study participants told us that this had introduced complexity and confusion surrounding cervical screening commissioning, and they expressed concern that screening rates would decline as a result, with some areas potentially affected more than others due to differences in local contextual conditions. In order to explore this further, we designed a quantitative analysis to test the proposition that CCGs most affected by this increase in complexity would have a greater decline in screening rates. Based on the findings from our interviews that working with more than one local authority acted to increase the complexity associated with the commissioning role, we compared screening rates between those CCGs which relate to a single local authority and those required to work with two or more local authorities.

## Quantitative findings

There were 14.1 million women eligible for screening in England in 2016.[21] Cervical screening rates decreased over time and the decline predated the implementation of the HSCA in April 2013. Unassisted delivery rates also declined over time. The relative decline between the first year (2009–10) and the last year (2015–16) for unassisted deliveries (−4.17%) was larger than for cervical screening (−2.70%) (table 3).

The changes in cervical screening rates over time were similar for practices in CCGs dealing with a single local authority (−2.53%) compared with practices in CCGs working with multiple local authorities (−2.87%). Figure 1 illustrates the trends in rates of cervical screening in the preintervention and postintervention periods for CCGs depending on the number of local authorities they worked with. There is a noticeable and sharp decline in the rates in both groups between 2011–12 and 2012–13.

Comparing the unadjusted averages for the pre-HSCA and post-HSCA years, cervical screening rates decreased by 0.39% more for GP practices located in CCGs working with multiple local authorities compared with practices in CCGs working with a single local authority. Unassisted birth rates decreased by 0.40% less for GP practices in CCGs working with multiple local authorities compared with GP practices in CCGs working with a single local authority. As maternity services were largely unaffected by the HSCA, we assumed that these differential changes captured the unmeasured population influences that confound comparisons of the changes in the two groups of CCGs. Relative to the decreases in unassisted delivery rates, GP practices in CCGs working with multiple local authorities experienced a decrease in cervical screening rates of 0.79% compared with practices in CCGs working with a single local authority (table 4).

The results were qualitatively similar when we estimated the formal triple difference (for all years and 2011–12 onwards only) and lagged dependent variable regression models (table 5). The triple-difference estimates show

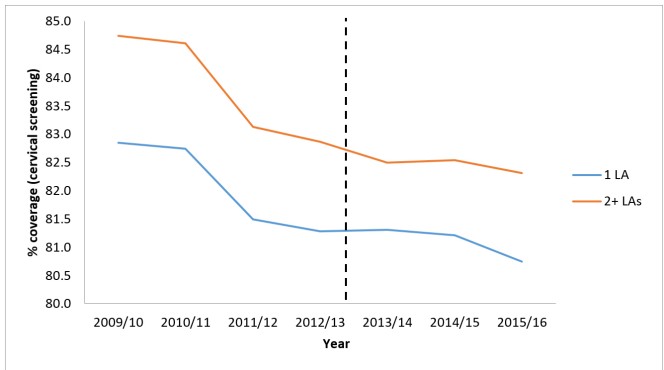

**Figure 1** Uptake (%) of cervical screening pre-HSCA and post-HSCA. HSCA, Health and Social Care Act 2012; LA, local authority.

**Table 4** Rates of cervical screening and unassisted birth for CCGs working with one and more than one LA, before and after the introduction of the HSCA

| Condition | CCGs working with one LA | | | CCGs working with more than one LA | | | Difference between change in CCGs working with more than one LA and change in CCGs working with one LA | Difference in changes for cervical screening minus difference in changes for unassisted births |
|---|---|---|---|---|---|---|---|---|
| | Average in the pre-HSCA years | Average in the post-HSCA years | *Change* | Average in the pre-HSCA years | Average in the post-HSCA years | *Change* | | |
| | Per cent | | | | | | | |
| Affected (cervical screening rates) | 82.09 | 81.09 | *−1.00* | 83.84 | 82.45 | *−1.39* | -0.39 | 0.78 |
| Unaffected (unassisted birth rates) | 62.80 | 60.82 | *−1.98* | 63.56 | 61.98 | *−1.58* | 0.40 | |

Values for pre and post are averages for all years in pre and post periods. The averages are weighted by the denominators used in the calculation of the rates. These are the eligible populations; the number of women aged between 25 and 64 years or the number of maternal deliveries.
CCG, Clinical Commissioning Group; HSCA, Health and Social Care Act 2012; LA, local authority.

that there was a differentially larger decline of 0.62% (95% CI −0.941 to −0.297) (model 1) in cervical screening rates for practices located in CCGs working with more than one local authority. The decrease is smaller using the shorter preperiod (0.259%; 95% CI −0.573 to 0.052, model 2).

The direction of result is robust to the model specification and, although we rejected the assumption of parallel trends for model 1 (all years), we could not reject the assumption for model 2 (2011–12 onwards). We also found a similar result in model 3 using the lagged dependent variable estimator, which yields unbiased estimates when pretrends cannot be assumed to be parallel.

The results are also robust to different groupings of the number of local authorities that CCGs work with. Table 5 includes model estimates comparing CCGs working with one or two local authorities with CCGs working with more than two local authorities. The direction of results is equivalent; and the scale and significance are either equivalent or increased. The same pattern is repeated in terms of tests of parallel trends. We cannot reject the null hypothesis of parallel trends for model 2 and the LDV estimation is preferable to model 1 in which we can reject the null hypothesis of parallel trends.

## DISCUSSION

We conducted a mixed methods study exploring the impact of changes associated with the HSCA in the English NHS on cervical screening rates. We carried out qualitative interviews with senior figures from a variety of relevant organisations in two large, socioeconomically diverse areas of England. Analysis of these interviews suggested that cervical screening commissioning had become more complex, with responsibilities between organisations less certain, as a consequence of the HSCA. Some interviewees predicted there would be a reduction in cervical screening rates in particular areas.

**Table 5** Triple-difference regression results

| | Difference-in-differences models | | | | Lagged dependent variable models | |
|---|---|---|---|---|---|---|
| | All years (1) | | 2011–12 onwards (2) | | All years (3) | |
| | Coefficient | 95% CI | Coefficient | 95% CI | Coefficient | 95% CI |
| Triple differences | −0.617 | −0.941 to −0.297 | −0.259 | −0.573 to 0.052 | −0.238 | −0.446 to −0.031 |
| Triple differences (alternative grouping of CCGs) | −0.774 | −1.134 to −0.420 | −0.440 | −0.786 to −0.099 | −0.234 | −0.461 to −0.009 |
| Number of observations | 105 745 | | 75 099 | | 44 472 | |
| Test of parallel trends | **F(3, 7672)** | **P value** | **F(1, 7667)** | **P value** | **N/A** | |
| | 4.89 | 0.002 | 0.33 | 0.567 | | |

Values are regression estimations from weighted least squares models on the empirical logit transformation of the rate including practice condition-specific fixed effects, full interaction of year with condition; and full interaction of year with the dummy for (N of LAs). Weighted by the denominators used to calculate the rates. Robust SEs, clustered by practice.
The triple difference represents (the change over time in cervical screening rates for CCGs working with only one LA minus the change over time in cervical screening rates for CCGs working with more than one LA) minus (the change over time in unassisted birth rates for CCGs working with only one LA minus the change over time in unassisted birth rates for CCGs working with more than one LA).
This model uses an alternative grouping of CCGs based on the number of LAs they work with ([1 or 2] vs [>2]).
LDV also contains values of the dependent variable in each of the pre intervention years. Estimated only on post intervention years.
CCG, Clinical Commissioning Group; LA, local authority; N/A, not available.

These findings prompted the development of an analysis to explore these issues quantitatively via a triple-difference regression analysis of publicly available data on cervical screening activity. To control for unmeasured confounders, we compared cervical screening rates with trends in unassisted birth rates because the commissioning of maternity services was unchanged pre-HSCA and post-HSCA.

Interviewees suggested a number of factors that might contribute to a reduction in cervical screening activity. Sexual health service commissioning responsibility had shifted to local authorities while NHSE was made responsible for commissioning national screening services, including cervical screening. Faced with financial austerity and cuts to their budgets, many local authorities were retendering their sexual health services with sexual health service providers but not including routine cervical screening. NHSE was also seemingly reluctant to commission sexual health clinics to perform cervical screening tests because this would entail a multitude of low-value contracts with numerous providers. This would be administratively laborious and practically difficult given the large size of NHSE's administrative areas and small numbers of NHSE commissioning staff in each area.

The quantitative analysis was designed to explore whether cervical screening activity had declined in areas most affected by commissioning organisational change. GP practices located in CCGs dealing with multiple local authorities, and therefore most exposed to increased commissioning complexity and potential disruption in services because of the lack of clarity of the roles of different organisations, experienced a larger decrease over time in cervical screening rates compared with practices in CCGs dealing with a single local authority. The opposite pattern was observed for unassisted births, which decreased more over time in the CCGs dealing with a single local authority. The triple-difference analyses confirmed that the effects were statistically significant and robust to different model specifications.

We have demonstrated unintended consequences arising out of a large-scale health system reform. Taken together, our findings suggest that there is an urgent need for clarification as to who holds the budget, and therefore who should be commissioning, cervical screening in the English NHS, and for local agreements to ensure that issues over funding and budgets do not disrupt screening programmes. More broadly, the issues we have identified in this study are of value to policy makers and system leaders in other health systems. The current study suggests that there are particular problems associated with service commissioning where coordination is required between multiple commissioners. This suggests that future commissioning reforms should include assessment of the likely impact on coordination, and a presumption in favour of commissioning all required services for geographical populations where possible. This may also have implications for mixed health systems, in which multiple payers (including public and private insurers as well as out of pocket payments) are responsible for services. In such systems achieving desirable population coverage for services such as screening may require specific coordination efforts.

## Potential confounders and study strengths

We took 2009 as our starting point for pre-HSCA cervical screening activity. Two potential confounders to our results were considered. First, the high-profile case of Jade Goody, a reality TV star who was diagnosed with cervical cancer in August 2008 and died in March 2009. The contemporaneous media attention and publicity was linked with a substantial increase in cervical screening rates (around an extra half a million women) during the time between Goody's diagnosis and death. However, previous impacts of high-profile cases of celebrity cancer diagnoses on population behaviour have tended to be brief and immediate rather than longer-lasting, and, therefore, we are confident that from 2010, rates of cervical screening returned towards underlying trends.[22] Second, the UK's Human Papillomavirus (HPV) vaccination programme was introduced in 2008 in order to reduce the incidence of cervical cancer.[23] The vaccine is offered to all girls aged 12–13 years, and figures for 2008–14 show high uptake rates of just under 90%. It is likely that this vaccination programme will contribute to a reduction in cervical screening activity in future. However, the first cohort of women in the programme, that is, those aged 12–13 years in 2008, were aged only 21–22 years in 2016–17, hence too young to have been invited for routine cervical screening (which begins at age 25) at the time of the study. We can, therefore, be confident that any changes to cervical screening rates cannot yet be attributed directly to the HPV programme, but any future research into cervical screening rates needs to take this into account.

We considered whether the results were sensitive to the group of CCGs in terms of the number of local authorities they worked with. The direction of results was the same, and the strength and significance was increased, comparing CCGs working with one or two local authorities with those CCGs working with more than two local authorities. We also considered whether the results were sensitive to the choice of comparison indicator (unassisted births) for maternity services. We tested whether the results would hold for another indicator of maternity services: the rate of deliveries by caesarean section. We observed the same direction and significance of results for this indicator as well.

The average age of mothers at delivery is likely to be younger than the average age of women attending for cervical screening. For our analysis, we require that differential changes in maternity indicators between CCGs with simple and CCGs with complex local authority relationships are a good proxy for other factors influencing cervical screening rates. We have confirmed the empirical validity of this assumption by looking for parallel trends in

the period before the HSCA, but we can never be entirely sure of its validity.

The findings presented here come from a longitudinal study of major healthcare system reform conducted by a multidisciplinary research team. The nature of this study facilitated the development of the relatively novel, sequential mixed methods approach in which the claims made in qualitative interviews could be tested in a subsequent quantitative analysis. There is a reinforcing effect in this analytical approach, which provides a strong cumulative indication that in areas of the country where complexity and coordination issues linked to the HSCA were more likely to occur there was an associated reduction in cervical screening rates.

**Author affiliations**
¹Division of Population Health, Health Services Research, and Primary Care, University of Manchester, Manchester, UK
²School of Health Sciences, University of Manchester, Manchester, UK
³Faculty of Biology, Medicine and Health, University of Manchester, Manchester, UK
⁴Manchester Centre for Health Economics, University of Manchester, Manchester, UK
⁵Division of Nursing, Midwifery and Social Work, University of Manchester, Manchester, UK
⁶Public Health and Policy, London School of Hygiene and Tropical Medicine, London, UK
⁷Health Services Research Unit, London, UK

**Acknowledgements** The authors would like to thank the research participants for their involvement, and acknowledge the valuable advice of the Project Advisory Group.

**Contributors** KC designed the study with input from MS, NM, PA, AC. JH, AH, LW-G gathered and analysed the qualitative data. TM and MS designed the quantitative evaluation and conducted this analysis. JH drafted the manuscript to which all authors made substantial contributions. All authors approved the final version and agree to be accountable for all aspects of the analysis.

**Funding** The report is based on independent research commissioned and funded by the NIHR Policy Research Programme ('Understanding the new commissioning system in England: contexts, mechanisms and outcomes', PR-R6-1113-25001).

**Disclaimer** The views expressed in the publication are those of the author(s) and not necessarily those of the NHS, the NIHR, the Department of Health, arm's-length bodies or other government departments.

**Competing interests** None declared.

**Patient consent for publication** Not required.

**Ethics approval** Ethical approval was granted by one of The University of Manchester Research Ethics Committees (application 15085) in March 2015. Participants were provided written information about the study, provided written consent or gave consent verbally at the beginning of telephone interviews.

**Provenance and peer review** Not commissioned; externally peer reviewed.

**Data sharing statement** No additional data are available.

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
