## [Reviewer comments · BMJ Open]

ARTICLE DETAILS

TITLE (PROVISIONAL)	Exploring the impacts of the 2012 Health and Social Care Act reforms to commissioning upon clinical activity in the English NHS: A mixed methods study of cervical screening
AUTHORS	Hammond, Jonathan; Mason, Thomas; Sutton, Matt; Hall, Alex; Mays, Nicholas; Coleman, Anna; Allen, Pauline; Warwick-Giles, Lynsey; Checkland, Kath

VERSION 1 - REVIEW

REVIEWER	Peter Watson University of Cambridge UK
REVIEW RETURNED	03-Oct-2018

GENERAL COMMENTS	Exploring the impacts of the 2012 Health and Social Care Act reforms to commissioning upon clinical activity in the English NHS: A mixed methods study of cervical screening bmjopen-2018-024156 I make one substantive point about modelling rates with other points relating to simplifying the description of the regression approach. The outcome variable appears to be a rate/percentage (e.g. page 8, line 53) expressed as a percentage such as that in Figure 1 on page 15. An ordinary least squares regression (page 9, line 52) is used to evaluate differences in these rates. If my above understanding is correct then a more appropriate approach to the linear regression used here would be either beta regression, a binomial or Poisson General Linear Model or modelling using the arcsine transform. This is because rates are bounded responses between 0% and 100% but the linear regression model being fitted isn't bounded. The linear regression model also incorrectly assumes that the variance of the outcome (rate) is independent of the rate itself which is incorrect since rates nearer 0% or 100% will have smaller variance than those around 50%. Page 9, lines 40-43. I am not clear here how the regression equation which is one for main effects and interactions of various factors is used to obtain the triple differences given in Table 5 on page 18 as implied by line 29 on page 9. Is the triple difference corresponding to a set or sets of regression coefficients in this regression model? It is not stated here what the outcome is other than it is a 'value' (page 9, line 45). I think you
---

	could express the model more clearly possibly even without reference to dummy variables since, the regression described here has main effects of practice, intervention, time and lagged values and two-way interactions involving these factors as predictors of rate. I think spelling it out algebraically gains nothing here as I would expect the readers to already be au fait with interactions, main effects and the like. Page 9, line 46. Should there be T-1 dummies for years rather than T as stated? The usual parameterisation e.g. in AN(C)OVAs is to have T-1 dummies ie one less than the number of levels of the factor to avoid linear dependencies which preclude least squares estimation. Page 9, line 52. I wondered if the regression should be described as a WEIGHTED least squares regression as stated in a footnote to Table 5 on page 18, line 35. Not clear what denominator refers to here on page 18, line 36 and in the footnote to Table 3 on page 14. Page 9, lines 53-54. I don't understand what you mean by clustering standard errors. Is this a pooling or using GP practice as a random/fixed effect? Page 17, Table 4. Are the estimates of rates here obtained by a weighted linear regression such as those in Table 5 on page 18 since both tables are estimating a triple difference in differences. (Regression is mentioned in a footnote to Table 5 but not in the one for Table 4). If not why would two approaches be used - namely a regression approach and another approach to measure the same outcome (of a triple difference in differences)? Page 18, lines 29-32 in Table 5. I assume the F test of parallel trends is associated with the testing of an interaction in the regression? This could be mentioned in the methods section of the paper when the regression is first mentioned. Page 18, Table 5. I wondered, as in Table 4 on page 17, if it might be more informative to decompose the triple difference into the sets of changes being differenced which yield the triple difference. ie stating the screening rates and their difference for one LA and multiple LAs for both cervical screening and unassisted births.
--	---

REVIEWER	David Candon University of Edinburgh, United Kingdom
REVIEW RETURNED	05-Oct-2018

GENERAL COMMENTS	MAJOR POINTS 1. My opinions on the mixed methods strategy are somewhat nuanced. On the one hand I am impressed by the way in which cervical screening wasn't part of the initial project design and only came up because of the qualitative research. It allays any fears that the readers would have that this may have been a finding which was data mined.
--

On the other hand, I'm somewhat concerned by the extent that you seem to think that the mixed methods approach helps with the causal inference. Typically for causal inference you need a mechanism for how the X variable is supposed to affect the Y variable (so we know the direction of causation) and then some sort of exogenous variation in the X variable (so we know what happens when only X changes). In this case I'm sceptical of the extent to which the qualitative analysis helps in determining the mechanism. It would be my prior belief that organizations with more responsibilities would fare worse than organizations with fewer responsibilities after a major reform. I don't need to read these interviews to be convinced of that. The article even contains a statement on page 5, lines 16/20, which gets this point across in a concise way "[Table 1] illustrates how responsibility for such services, previously commissioned by PCTs, are now split across different agencies. This increased complexity and fragmentation of responsibilities has the potential to disrupt service commissioning". To that end, the qualitative part of the analysis seems superfluous.

So, apart from pointing you in the direction of cervical screening (which I admit is an important point), is there anything else that qualitative analysis can add towards understanding the mechanism that I wouldn't have gotten from the above statement? Because if there isn't anything then pages 10-12 may be more suitable for an appendix. However, I am willing to accept that my issue with this may simply be due to a difference in disciplines (qualitative analysis is not used very often in mine).

2. I could be mistaken but I can't seem to find anything in the qualitative analysis which points towards why the econometric model was set up comparing CCGs working with one LA with CCGs working with multiple LAs. This could always be justified with economic theory (as I said above this is what my prior belief would be) but then this would undermine the extent to which the qualitative analysis is informing the quantitative analysis. Did the qualitative analysis play any role in this?

MINOR POINTS

Abstract

Page 2, Line 14: This should be difference-in-difference-in-differences or triple differences. Triple difference-in-differences doesn't sound correct, unless it is something different from what I am imagining? This appears in the main body of the article too. This could be cleared up by seeing the exact regression equation (see point on this below).

Page 2, Line 44/45: This sentence needs to be rewritten as it doesn't make sense as it is. I think "Minimize" should "minimizing". Or maybe something like "It is important to minimize the ambiguity over commissioning responsibilities that stems from large scale health system reform" would be more appropriate.

Methods

Page 8, Line 33: Could you present any data for how similar the women who have unassisted deliveries are to the women who go for cervical screening? For example, I'm inclined to think that the women who go for screening are much older considering their age group is 25 – 64. I'm sure the age for unassisted deliveries starts earlier and doesn't go up to 64. It would be illuminating to see some information on the mean ages of both groups. Even if it is

	not from the same data source, it could be useful to use some other NHS data to get a sense of how comparable the groups are. I think the UD group are more likely to capture the unobserved changes that could affect the CS group if they are more alike to begin with. If you can't get this then it is something that I would mention in the paper. Page 8, Line 40/46: While I appreciate the detail with which the authors have described the model in words I still think it would be useful to write out the econometric model explicitly. It will then make the comparison to the second model easier for readers to understand. Page 9, Line 26/27: I think you mean the estimator is less biased. It doesn't make sense to talk about the estimates being biased. Results Page 10, Line 7: Do you have some data on the amount of times these two themes were mentioned in relation to other themes? It would be interesting to know if they were by far the two most dominant themes or if they were, for example, the two most popular out of four very popular themes.
--	---

REVIEWER	Rhiannon Parker University of New South Wales
REVIEW RETURNED	19-Oct-2018

GENERAL COMMENTS	This is an impressive paper on an important topic. I wish to congratulate the authors on the quality of the design and data that have managed to obtain for both the Quantitative and Qualitative components of this paper. Although I generally think this paper is excellent I have a few comments that I think might make the manuscript better. I've arranged them by section. INTRODUCTION 1. In the intro, there are several sections that could benefit from some additional explanation. For example, what does centrally-driven mean? 2. Likewise, there are a large number of acronyms that required me to flick back and forward to remember what they all meant. The number of acronyms places a considerable burden on the reader. Is there any way these can be reduced? METHODS 3. What is meant by the phrase 'the early phases of thematic analysis'? More detail on the qualitative process would have been useful here. 4. It was not clear how ethics for the qualitative data was ensured. How was participant content achieved? 5. The quantitative design is sophisticated and impressive. However, it was difficult to tell what the 'treatment' and 'control' groups really were. I think this is in part due to the complexity of the reforms and the number of acronyms. Thus, I think a clear statement of what the treatment was and who was in which condition would be useful. RESULTS
---

	6. I found it unusual that the qual preceded the quant. While the authors are certainly in their rights to do so, I think the connection between the qual and the quant results would be enhanced by switching the order of the results. In addition, a section devoted to connecting the two results may be useful. For the qual could the authors explain the participant labeling schema? 7. The first theme title in the qual is broad and lacks clarity. 'Differing perspectives' is not really a theme, it's what will inevitably happen in interviews. I think a clearer description (e.g. 'confusion and uncertainty') would also make implications clear, namely, that clarity needs to be provided around organisational budgets and responsibilities. DISCUSSION 8. This article is very specific to the NHS and given this is the BMJ Open this likely makes sense. However, for international audiences it is important to provide readers with an understanding of what the general lessons from this study are that may apply to their health system.
--	--

VERSION 1 – AUTHOR RESPONSE

Thank you very much for these considered and helpful comments. We have addressed these in detail in the table below.

	R1	Response
1	The outcome variable appears to be a rate/percentage (e.g. page 8, line 53) expressed as a percentage such as that in Figure 1 on page 15. An ordinary least squares regression (page 9, line 52) is used to evaluate differences in these rates. If my above understanding is correct then a more appropriate approach to the linear regression used here would be either beta regression, a binomial or Poisson General Linear Model or modelling using the arcsine transform. This is because rates are bounded responses between 0% and 100% but the linear regression model being fitted isn't bounded. The linear regression model also incorrectly assumes that the variance of the outcome (rate) is independent of the rate itself which is incorrect since rates nearer 0% or 100% will have smaller variance than those around 50%.	We have re-estimated the regression models using the empirical logit transformation (Stevens S et al. Analysing indicators of performance, satisfaction, or safety using empirical logit transformation. BMJ 2016;352:i1114). This approach is most suitable, given the requirement for weighted regression and a large number of fixed effects. These results now appear in Table 5 and the method is explained on pages 10-11.
2	Page 9, lines 40-43. I am not clear here how the regression equation which is one for main effects and interactions of various factors is used to obtain the triple differences given in Table 5 on page 18 as implied	The coefficient of interest is the ρ coefficient on the $L_{j\pi c}$ term, but the referee is correct that we did not say so explicitly. The referee makes an important point about whether the algebra is needed and the effect on the accessibility of the material for a general audience.

	by line 29 on page 9. Is the triple difference corresponding to a set or sets of regression coefficients in this regression model? It is not stated here what the outcome is other than it is a 'value' (page 9, line 45). I think you could express the model more clearly possibly even without reference to dummy variables since, the regression described here has main effects of practice, intervention, time and lagged values and two-way interactions involving these factors as predictors of rate. I think spelling it out algebraically gains nothing here as I would expect the readers to already be au fait with interactions, main effects and the like.	We have therefore moved the algebra to an appendix and have summarised the estimation approach in the text (p.10).
3	Page 9, line 46. Should there be T-1 dummies for years rather than T as stated? The usual parameterisation e.g. in AN(C)OVAs is to have T-1 dummies ie one less than the number of levels of the factor to avoid linear dependencies which preclude least squares estimation.	This has been corrected in the material that is now in the technical appendix.
4	Page 9, line 52. I wondered if the regression should be described as a WEIGHTED least squares regression as stated in a footnote to Table 5 on page 18, line 35. Not clear what denominator refers to here on page 18, line 36 and in the footnote to Table 3 on page 14.	We have added "weighted" to the description of the estimated models (page 10). The weights are the denominators used in the calculation of the rates. These are the eligible populations; the number of women aged between 25 and 64 years or the number of maternal deliveries. This has been added to the footnotes of Tables 3 and 5.
5	Page 9, lines 53-54. I don't understand what you mean by clustering standard errors. Is this a pooling or using GP practice as a random/fixed effect?	The standard errors have been adjusted for dependence within GP practice using the cluster() option in Stata. This was first described by Rogers (1993). We have added that reference and clarification.
6	Page 17, Table 4. Are the estimates of rates here obtained by a weighted linear regression such as those in Table 5 on page 18 since both tables are estimating a triple difference in differences. (Regression is mentioned in a footnote to Table 5 but not in the one for Table 4). If not why would two approaches be used - namely a regression approach and another approach to measure the same outcome (of a triple difference in differences)?	Table 4 presents unadjusted summary statistics for the rates and the changes in the rates. Table 5 contains the results of the regression analyses. We have clarified this distinction in the text and in the titles and notes of the tables. In particular, we no longer refer to the final column of Table 4 as a triple-difference.
7	Page 18, lines 29-32 in Table 5. I assume the F test of parallel trends is associated with the testing of an interaction in the regression? This could be mentioned in the methods	We had described the analyses we undertook to assess the plausibility of the parallel trends assumption in the methods section of the original manuscript, but we had not described this as an F-test. We have now sharpened up the description of

	section of the paper when the regression is first mentioned.	the parallel trends testing in the methods section and explicitly mentioned the F-test presented in Table 5.
8	Page 18, Table 5. I wondered, as in Table 4 on page 17, if it might be more informative to decompose the triple difference into the sets of changes being differenced which yield the triple difference. Ie stating the screening rates and their difference for one LA and multiple LAs for both cervical screening and unassisted births.	We agree, and these are the figures that are presented in Table 4. We hope that the relabelling of the tables and the further explanations make clearer the role of Table 4 in supporting the analysis in the way the reviewer suggests.
	R2	
1	My opinions on the mixed methods strategy are somewhat nuanced. On the one hand I am impressed by the way in which cervical screening wasn't part of the initial project design and only came up because of the qualitative research. It allays any fears that the readers would have that this may have been a finding which was data mined. On the other hand, I'm somewhat concerned by the extent that you seem to think that the mixed methods approach helps with the causal inference. Typically for causal inference you need a mechanism for how the X variable is supposed to affect the Y variable (so we know the direction of causation) and then some sort of exogenous variation in the X variable (so we know what happens when only X changes). In this case I'm sceptical of the extent to which the qualitative analysis helps in determining the mechanism. It would be my prior belief that organizations with more responsibilities would fare worse than organizations with fewer responsibilities after a major reform. I don't need to read these interviews to be convinced of that. The article even contains a statement on page 5, lines 16/20, which gets this point across in a concise way "[Table 1] illustrates how responsibility for such services, previously commissioned by PCTs, are now split across different agencies. This increased complexity and fragmentation of responsibilities has the potential to disrupt service commissioning". To that end, the qualitative part of the analysis seems superfluous. So, apart from pointing you in the direction of cervical screening (which I	Thank you for this helpful observation. We realise that we had not fully elaborated on the inter-connections between the qual and quant work in our previous draft and have addressed this. We have added a section at the start of the qual results section (p.11) which explains that interviewees from CCGs working with multiple local authorities post-HSCA identified a number of challenges arising from this situation. This finding informed our assumption that organisations which must coordinate with a larger number of other organisations would fare worse than those with simpler inter-organisational arrangements (and we have also added additional content on p.8 to make this clear)—i.e. the mechanism. Notably, Fernandez et al. have recently adopted the same approach (https://www.sciencedirect.com/science/article/pii/S0167629618301000). The qualitative findings suggested that responsibilities around commissioning cervical screening were less clear post-HSCA and that there might be considerable variability between local authorities in terms of what cervical screening services, if any, they were commissioning (particularly as re-procurement of contracts occurred). Thus the more local authorities a CCG had to work with, the more difficult the task of commissioning sexual health services, including cervical screening. We disagree that the qualitative data and analysis might be moved to an appendix. The value of the qualitative work presented lies both in the identification of a potential issue to be further investigated quantitatively and in exploring the consequences (to cervical screening activity) of large scale health system reform resulting from national policy. The qualitative work provides an insight into the life-worlds of those working within the health service (including experiences of confusion and disruption) and the interplay between their work and the changes to screening resulting from the HSCA. We strongly believe that this context, texture, and depth is valuable because it helps readers to better understand the change associated with the policy,

	admit is an important point), is there anything else that qualitative analysis can add towards understanding the mechanism that I wouldn't have gotten from the above statement? Because if there isn't anything then pages 10-12 may be more suitable for an appendix. However, I am willing to accept that my issue with this may simply be due to a difference in disciplines (qualitative analysis is not used very often in mine).	and this is important if the issues that we raise are to be addressed by policy makers. Finally, in relation to the point regarding Table 1, we cited work from Peckham and colleagues that was published in 2017, towards the end of our period of fieldwork. This work was therefore not available at the time that we were conducting our study, and hence it was our own qualitative work that influenced our approach to further quantitative exploration.
2	I could be mistaken but I can't seem to find anything in the qualitative analysis which points towards why the econometric model was set up comparing CCGs working with one LA with CCGs working with multiple LAs. This could always be justified with economic theory (as I said above this is what my prior belief would be) but then this would undermine the extent to which the qualitative analysis is informing the quantitative analysis. Did the qualitative analysis play any role in this?	Please see previous point. While no doubt such an analysis could have been justified a priori on theoretical grounds (https://www.sciencedirect.com/science/article/pii/S0167629618301000), the decision to focus the comparative analysis in this particular way was strengthened by the fact that it also emerged strongly from the interview and other qualitative data.
3	Page 2, Line 14: This should be difference-in-difference-in-differences or triple differences. Triple difference-in-differences doesn't sound correct, unless it is something different from what I am imagining? This appears in the main body of the article too. This could be cleared up by seeing the exact regression equation (see point on this below).	We have changed the description to triple-difference throughout.
4	Page 2, Line 44/45: This sentence needs to be rewritten as it doesn't make sense as it is. I think "Minimize" should "minimizing". Or maybe something like "It is important to minimize the ambiguity over commissioning responsibilities that stems from large scale health system reform" would be more appropriate.	We have corrected this.
5	Page 8, Line 33: Could you present any data for how similar the women who have unassisted deliveries are to the women who go for cervical screening? For example, I'm inclined to think that the women who go for screening are much older considering their age group is 25 – 64. I'm sure the age for unassisted deliveries starts earlier and doesn't go up to 64. It would be illuminating to see some information on the mean ages of both groups. Even if it is not from the same data source, it could be useful to use some other NHS data to get a sense of how comparable the groups are. I think the UD group	The latest edition of NHS Maternity Statistics indicates that 82% of mothers at delivery are aged 25 years or older and 78% are aged 25-39 years. Reference: https://digital.nhs.uk/data-and-information/publications/statistical/nhs-maternity-statistics/2016-17 Coverage rates of the cervical screening programme are available for broad age groups. Reference: https://digital.nhs.uk/data-and-information/data-tools-and-services/data-services/general-practice-data-hub/cervical-screening-programme-coverage Target frequency is higher for women aged 25 to 49 years (every 3 years) than for women aged 50 to 64 years (every 5 years).

	are more likely to capture the unobserved changes that could affect the CS group if they are more alike to begin with. If you can't get this then it is something that I would mention in the paper.	Although difficult to get comparable statistics, the reviewer is likely to be correct that the average age of mothers at unassisted deliveries is lower than the average age of women attending for cervical screening. The key assumption for our analysis is that differential changes in the unassisted delivery rate between CCGs with simple and CCGs with complex LA relationships are a proxy for the unobserved changes. We test this assumption using the parallel trends test, but we can never be sure of the validity of this control group. We have added explicit mention of this issue in the discussion section (p. 23).
6	Page 8, Line 40/46: While I appreciate the detail with which the authors have described the model in words I still think it would be useful to write out the econometric model explicitly. It will then make the comparison to the second model easier for readers to understand.	To reach the widest possible audience, we believe that there is value in describing the econometric analysis in words and in algebraic terms. The views of this reviewer and reviewer 1 seem to support this approach. We have therefore retained the description of the analyses in words in the main text and have created a technical appendix for the algebraic presentations of the models.
7	Page 9, Line 26/27: I think you mean the estimator is less biased. It doesn't make sense to talk about the estimates being biased.	We have changed the text to read "...is a less biased estimator of treatment effects..." (p.10).
8	Page 10, Line 7: Do you have some data on the amount of times these two themes were mentioned in relation to other themes? It would be interesting to know if they were by far the two most dominant themes or if they were, for example, the two most popular out of four very popular themes.	We have clarified that the two themes presented are those relating directly to issues surrounding cervical screening post-Act. Their inclusion is not due to the particular volume of data coded within them (they are two of many themes relating to screening and sexual health service commissioning more broadly) but rather their value as an organising device for helping to explain the issues around cervical screening which we identified as important.
R3		
1	This is an impressive paper on an important topic. I wish to congratulate the authors on the quality of the design and data that have managed to obtain for both the Quantitative and Qualitative components of this paper. Although I generally think this paper is excellent I have a few comments that I think might make the manuscript better. I've arranged them by section.	Thank you.
2	In the intro, there are several sections that could benefit from some additional explanation. For example, what does centrally-driven mean?	We have made this clearer.
3	Likewise, there are a large number of acronyms that required me to flick back and forward to remember what they all meant. The number of acronyms places a considerable burden on the	We appreciate that too many acronyms can be burdensome for the reader. We have reduced the number somewhat but are aware that doing so further might add an unacceptable amount of additional length to the paper.

	reader. Is there any way these can be reduced?	
4	What is meant by the phrase 'the early phases of thematic analysis'? More detail on the qualitative process would have been useful here.	We have expanded this section slightly to be clearer about the process of qualitative analysis and the production of themes (p.8).
5	It was not clear how ethics for the qualitative data was ensured. How was participant content achieved?	We have added information about ethics approval and participant consent to the footnote section.
6	The quantitative design is sophisticated and impressive. However, it was difficult to tell what the 'treatment' and 'control' groups really were. I think this is in part due to the complexity of the reforms and the number of acronyms. Thus, I think a clear statement of what the treatment was and who was in which condition would be useful.	We hope that the revised presentation of the quantitative design clarifies this point.
7	I found it unusual that the qual preceded the quant. While the authors are certainly in their rights to do so, I think the connection between the qual and the quant results would be enhanced by switching the order of the results. In addition, a section devoted to connecting the two results may be useful. For the qual could the authors explain the participant labeling schema?	We believe that switching the order of the qualitative and quantitative findings in the paper would be confusing since the qualitative data collection preceded the quantitative analysis, and was deliberately designed, in part, to contribute to the development of hypotheses and a focus for the ensuing quantitative analysis. This approach was undertaken specifically because we wanted to look at the quantitative impacts of the HSCA reforms of the commissioning system, yet, at the outset of the research had no straightforward way of doing so, given that the reforms were introduced across England at the same time. There was no obvious or feasible comparison population or system we would use to generate a quasi-experiment until we had undertaken some initial qualitative data collection. This initial qualitative phase began to enable us to identify that the degree of disruption produced by the reforms differed between services (sexual health services being particularly affected) and areas (some CCGs were faced with new, more complex inter-organisational relationships with local authorities than others). This enabled us to begin to identify possible ways of modelling quantitatively the impact of the reforms in services and areas more and less likely to be affected by the changes. Regarding a section linking the qual and quant results: at the start of the qualitative findings section we have explained how interviewees from CCGs working with multiple local authorities post-Act experienced a number of challenges, which helps to inform how and why we developed our particular quantitative approach. We have also added a short summary at the end of this section to summarise how the qualitative findings informed the qualitative work. We believe this makes the connection much clearer. The schema for interview data extracts has been added on p.11.
8	The first theme title in the qual is broad and lacks clarity. 'Differing	We agree and have updated the title of the theme accordingly.

	perspectives' is not really a theme, it's what will inevitably happen in interviews. I think a clearer description (e.g. 'confusion and uncertainty') would also make implications clear, namely, that clarity needs to be provided around organisational budgets and responsibilities.	
9	This article is very specific to the NHS and given this is the BMJ Open this likely makes sense. However, for international audiences it is important to provide readers with an understanding of what the general lessons from this study are that may apply to their health system.	We have added a section on page 22 to do exactly this.

VERSION 2 – REVIEW

REVIEWER	Peter Watson MRC Cognition and Brain Sciences Unit Cambridge UK
REVIEW RETURNED	12-Dec-2018

GENERAL COMMENTS	Exploring the impacts of the 2012 Health and Social Care Act reforms to commissioning upon clinical activity in the English NHS: A mixed methods study of cervical screening bmjopen-2018-024156.R1 The authors have now mentioned that they have used a logistic regression in their analysis of rates (page 11, lines 59-60 and on page 21 in Table 5, line 32) which is a correct analysis. Some points of clarity: I notice that the logistic regression is described as a WEIGHTED least squares regression (page 11, line 54) rather than a logistic regression. Page 21, line 32 mentions the weights are a total (denominator). Not sure though from this what this is a total of (populations of people in each practice?) as only the word 'denominator' is mentioned. Is this weighted analysis just another way of describing the binary logistic regression since you can specify totals when performing it if using aggregated data? I also notice in Table 5 (page 21, line 32) a mention of robust standard errors clustered by practice. This is stated again elsewhere (page 12, line 4) but neither mention makes it clear how you clustered by practice. Be good to have a bit more on this (e.g. on page 12, line 4) especially as the reference for the clustering given on page 12, line 4 (STATA technical bulletin rather than a journal) strikes me as somewhat obscure and may be difficult for non-STATA users to access and use. Is this clustering simply resulting from treating practice as a fixed effect e.g. inputting dummy regressors for practice (as implied on lines 55-56 on page 11 and also in the technical appendix)?
---

	Did you consider treating practice as a random factor so removing its effect using a variance component? If there are a lot of practices and they are regarded as nuisance variables so not of interest (as here) it often makes sense to treat practice effect as a single random effect and fit a single variance component term to estimate across practice variance. Similarly when you mention a robust standard error is this a specially worked out standard error (e.g. Huber-White?) or are the standard errors considered robust simply due to the removal of practice effects? I assume you are providing annotated STATA code for this paper in the spirit of Open Science (possibly via an appendix) in order for readers to fit the logistic regression models used in this paper? It would also be useful to specify which STATA procedure was used for the logistic regressions in the text e.g. on page 11, line 54 where you mention the use of STATA.
--	--

REVIEWER	David Candon University of Edinburgh
REVIEW RETURNED	14-Dec-2018

GENERAL COMMENTS	I think the authors have done an excellent job of addressing my comments, as well as those of the other reviewers. I especially like the addition of the original regression to the Appendix. It offers a very concise description of what exactly the model is. I have no further comments to make.
--

VERSION 2 – AUTHOR RESPONSE

Thank you very much for these further comments. Our responses are provided in the table below.

	R1	Response
1	"The authors have now mentioned that they have used a logistic regression in their analysis of rates (page 11, lines 59-60 and on page 21 in Table 5, line 32) which is a correct analysis. Some points of clarity: I notice that the logistic regression is described as a WEIGHTED least squares regression (page 11, line 54) rather than a logistic regression."	We have not used logistic regression to estimate the models. Instead we have used the empirical logit transformation as described by Stevens et al (BMJ 2016;352:i1114). The regression models have been estimated using weighted least squares regression on the empirical logit transformation of the rate.
2	"Page 21, line 32 mentions the weights are a total (denominator). Not sure though from this what this is a total of (populations of people in each practice?) as only the word 'denominator' is mentioned. Is this weighted analysis just another way of describing the binary logistic regression since you can specify totals when performing it if using aggregated data?"	The weights are the eligible populations. The denominator for the cervical screening rate is the number of women aged between 25 and 64 years registered with the practice. The denominator for the unassisted birth rate is the number of maternal deliveries by women registered with the practice. We weight the analyses by the denominators to allow for greater accuracy in rates estimated over larger populations or numbers of events.

3	“I also notice in Table 5 (page 21, line 32) a mention of robust standard errors clustered by practice. This is stated again elsewhere (page 12, line 4) but neither mention makes it clear how you clustered by practice. Be good to have a bit more on this (e.g. on page 12, line 4) especially as the reference for the clustering given on page 12, line 4 (STATA technical bulletin rather than a journal) strikes me as somewhat obscure and may be difficult for non-STATA users to access and use. Is this clustering simply resulting from treating practice as a fixed effect e.g. inputting dummy regressors for practice (as implied on lines 55-56 on page 11 and also in the technical appendix)? Did you consider treating practice as a random factor so removing its effect using a variance component? If there are a lot of practices and they are regarded as nuisance variables so not of interest (as here) it often makes sense to treat practice effect as a single random effect and fit a single variance component term to estimate across practice variance. Similarly when you mention a robust standard error is this specially worked out standard error (e.g. Huber-White?) or are the standard errors considered robust simply due to the removal of practice effects?”	We have included fixed effects (dummy variables) for practices and we have clustered the standard errors at practice level. We believe that this is standard in analysis of this type (see, for example, Woolridge (2012), and Angrist and Pischke (2013)), as it allows for different mean values across practices and correlation in the errors within practices. References: Angrist, J. D., & Pischke, J. S. (2013). Mostly harmless econometrics: an empiricists companion. Cram101 Publishing. Wooldridge, J. M. (2012). Introductory Econometrics: A Modern Approach. Boston: Cengage Learning.
4	“I assume you are providing annotated STATA code for this paper in the spirit of Open Science (possibly via an appendix) in order for readers to fit the logistic regression models used in this paper? It would also be useful to specify which STATA procedure was used for the logistic regressions in the text e.g. on page 11, line 54 where you mention the use of STATA.”	The procedure that we have followed to obtain the empirical logit transformation of the proportion is very nicely explained by Stevens et al (BMJ, 2016) and we do not think that this needs to be repeated in this paper. We have, however, included the STATA syntax in the text, as suggested (p11): “The general form of the STATA command is: areg {depvar} {indepvars} [aw=denom], robust absorb(practicexindicator) cluster(practice).”
R2		
1	“I think the authors have done an excellent job of addressing my comments, as well as those of the other reviewers. I especially like the addition of the original regression to the Appendix. It offers a very concise description of what exactly the model is. I have no further comments to make.”	Thank you very much.